# Depletion of IGFALS Serum Level up to 3 Months After Cardiac Surgery, with Exploration of Potential Relationships to Surrogates of Organ Failures and Clinical Outcomes

**DOI:** 10.3390/cimb47080581

**Published:** 2025-07-23

**Authors:** Krzysztof Laudanski, Mohamed A. Mahmoud, Hossam Gad, Daniel A. Diedrich

**Affiliations:** 1Department of Anesthesiology and Perioperative Care, Mayo Clinic, Rochester, MN 55905, USA; gad.hossam@mayo.edu (H.G.); diedrich.daniel@mayo.edu (D.A.D.); 2Division of Pulmonology and Critical Care, Mayo Clinic, Rochester, MN 55905, USA; mohamed.kh.mogge@gmail.com

**Keywords:** cardiac surgery, insulin-like growth factor binding protein acid-labile subunit, IGFALS, perioperative inflammation, acetaminophen, acetylsalicylic acid

## Abstract

The insulin-like growth factor binding protein, acid-labile subunit (IGFALS), plays a crucial role in glucose metabolism and immune regulation, key processes in recovery from surgery. Here, we studied the perioperative serum IGFALS dynamics and explored potential clinical implications. A total of 79 patients undergoing elective cardiac surgery with implementation of cardiopulmonary bypass had their serum isolated at baseline, 24 h, seven days, and three months postoperatively to assess serum concentrations of IGFALS and insulin growth factor 1 (IGF-1). Markers of perioperative injury included troponin I (TnI), high-mobility group box 1 (HMGB-1), and heat shock protein 60 (Hsp-60). Inflammatory status was assessed via interleukin-6 (IL-6) and interleukin-8 (IL-8). Additionally, we measured in vitro cytokine production to viral stimulation of whole blood and monocytes. Surrogates of neuronal distress included neurofilament light chain (NF-L), total tau (τ), phosphorylated tau at threonine 181 (τp181), and amyloid β40 and β42. Renal impairment was defined by RIFLE criteria. Cardiac dysfunction was denoted by serum N-terminal pro-brain natriuretic peptide (NT-proBNP) levels. Serum IGFALS levels declined significantly after surgery and remained depressed even at 3 months. Administration of acetaminophen and acetylsalicylic acid differentiated IGFALS levels at the 24 h postoperatively. Serum IGFALS 24 h post-operatively correlated with production of cytokines by leukocytes after in vitro viral stimulation. Serum amyloid-β1-42 was significantly associated with IGFALS at baseline and 24 h post-surgery Patients discharged home had higher IGFALS levels at 28 days and 3 months than those discharged to healthcare facilities or who died. These findings suggest that IGFALS may serve as a prognostic biomarker for recovery trajectory and postoperative outcomes in cardiac surgery patients.

## 1. Introduction

Insulin-like growth factor binding protein, acid-labile subunit (IGFALS), plays a critical role in glucose metabolism through its interactions with insulin and growth hormones [1,2]. Functionally, IGFALS is essential for regulating IGF via stabilizing the ternary complex comprising insulin-like growth factor (IGF), acid labile subunit (ALS), and insulin growth factor binding protein (IGFBP) 3 and 5 [2,3,4,5]. Beyond its metabolic functions, IGFALS contributes to bone growth, immune function, and neuropsychiatric health [6,7,8,9,10,11]. Unsurprisingly, both inherited and acquired deficiencies in IGFALS are associated with growth impairment and metabolic dysregulation [6,7,10,12,13,14,15,16]. The liver is the primary site of IGFALS synthesis, with some evidence indicating additional expression in the kidneys [17,18]. Only recently, Xu et al demonstrated that IGFALS modulates key immune signaling components, including TNF receptor-associated factor 6 (TRAF6) and IκB kinase (IKK), during viral challenges [11]. Conversely, IGFALS expression is tightly regulated by several factors affected by inflammation. Growth hormone (GH) enhances IGFALS mRNA and serum protein levels via transcription factors STAT5, a process facilitated by Janus kinase 2 signaling [19,20]. Conversely, inflammatory cytokines downregulate IGFALS expression via the suppressor of cytokine signaling-3 (SOCS3) pathway [21,22,23].

During critical illness, variability in IGFALS expression has been linked to disrupted GH signaling, with further modulation by sex-specific hormonal influences [24,25]. Emerging research highlights a potential role for IGFALS in regulating the inflammatory response and in influencing clinical outcomes in intensive care unit (ICU) settings [23,26,27,28,29]. Despite these findings, the role of IGFALS in acute sterile inflammation, such as that triggered by major surgery, remains poorly understood [11]. Cardiac surgery triggers a well-characterized sterile inflammatory and metabolic stress [30,31,32,33]. Dysregulation of IGFALS in this context may adversely impact glucose metabolism, vascular remodeling, immune activation, cellular proliferation, and programmed cell death [5,11,15,34,35,36]. So, in patients with underlying endocrine or inflammatory conditions, such as rheumatic heart disease, baseline IGFALS abnormalities are common [37].

Theoretically, perioperative IGFALs dynamics could affect surgical outcomes ranging from postoperative cardiac function, arrhythmias, and long-term neurocognitive recovery [38,39,40,41]. Accordingly, this study explored the time-wise dynamic of serum IGFALS levels in patients undergoing cardiac surgery and contrasted it to IGF. To investigate its immunomodulatory role, we evaluated correlations between serum IGFALS levels and immune responses to in vitro viral stimulation [11]. Finally, we assessed whether changes in IGFALS levels were associated with serum surrogates of cardiac and neuronal injury [42,43].

## 2. Materials and Methods

### 2.1. Study Cohort and Consent Process

A total of 79 patients were prospectively enrolled in the pilot study. Inclusion criteria required that patients be scheduled for non-emergent cardiac surgery, be capable of providing informed consent, and be adults. Exclusion criteria included lack of consent, emergent procedures, or treatment with immunosuppressive agents. All patients were recruited in accordance with a study protocol approved by the Institutional Review Board at the University of Pennsylvania (IRB# 815686). The study was conducted in full compliance with the ethical principles outlined in the Declaration of Helsinki.

### 2.2. Collection of the Electronic Data

Electronic medical records (EMRs) were reviewed to extract relevant demographic and clinical data. The Charlson Comorbidity Index (CCI) quantified the burden of pre-existing comorbidities [44]. The Acute Physiology and Chronic Health Evaluation II (APACHE II) score was calculated at 1, 24, and 48 h post-admission [45]. Organ dysfunctions were evaluated using the Marshall Organ Dysfunction Score (MODS) [46]. Preoperative cardiac function was assessed using serum troponin I (TnI) levels obtained prior to surgery, as documented in the EMR. Postoperative renal function was evaluated using serum creatinine values from routine laboratory testing, applying the RIFLE criteria to determine the presence and severity of acute kidney injury (AKI) [47]. Additionally, incidences of pulmonary embolism, deep venous thrombosis, and cerebrovascular events during the perioperative period were extracted from the EMR for outcome analysis.

### 2.3. Measurements of Biological Variables

Samples were collected preoperatively (t_baseline_), followed by 24 h postoperatively (t_24h_), seven days postoperatively (t_7d_), and three months into recovery (t_3m_). Venous blood was drawn into heparinized BD™ Vacutainer tubes and immediately processed by centrifugation at 2000× *g* for 5 min at 4 °C. Obtained serum was aliquoted and stored at −80 °C until further analysis.

Serum levels of IGFALS and IGF-1 were measured using commercially available ELISA kits (BioLegend, San Diego, CA, USA). Pro-inflammatory cytokines IL-6 and IL-8 were assessed using kits from ThermoFisher (Waltham, MA, USA). Alarmins, including Hsp-60 and HMGB-1, were quantified using Luminex-based assays (Luminex, Madison, WI, USA) and ELISA (Aviva, Auburn, MA, USA), respectively.

Markers of neurodegeneration—including total tau (τ), amyloid-β1-40, and amyloid-β1-42—were analyzed via Luminex technology (Luminex, Madison, WI, USA). Serum N-terminal pro–B-type natriuretic peptide (NT-proBNP) levels were measured using a bead-based assay (ThermoFisher, Waltham, MA, USA) and analyzed on a MagPix instrument (Luminex, Madison, WI, USA). Serum levels of growth hormone and IL-1β were assessed using ELISA kits from Sino Biological (Wayne, PA, USA) and BioLegend (San Diego, CA, USA), respectively.

To evaluate immune responsiveness by negatively isolated monocyte isolated as before, 10^5^ of cells were stimulated for 18 h with lipopolysaccharide (LPS, 50 ng/mL; Enzo, Farmingdale, NY, USA) or H3N2 influenza virus (1 µg/mL; BEI, Manassas, VA, USA) in X-Vivo™ 10 medium (Lonza, Basel, Switzerland) [32]. Alternatively, whole blood was stimulated instead of isolated MO as specified before [48]. Culture supernatants were harvested for IL-6 and TNF-α quantification via ELISA (ThermoFisher, Waltham, MA, USA).

Total RNA was extracted using a commercial kit (Zymo Research, Irvine, CA, USA) and submitted for RNA sequencing, which was performed by BGI Genomics (Beijing, China).

### 2.4. Statistical Analysis

The Shapiro–Wilk W test and distribution plots were employed to assess the normality of continuous variables. Parametric data are presented as mean ± standard deviation (SD) and were compared using the Student’s *t*-test for two-group comparisons or one-way ANOVA for comparisons involving more than two groups. Non-parametric data are reported as median (M_e_) with interquartile ranges (IQR) and were analyzed using the Mann–Whitney *U* test for two-group comparisons. For paired or dependent data, appropriate paired tests were used, including the paired *t*-test, Wilcoxon signed-rank test, or the Friedman rank test, depending on data distribution. The biserial correlation coefficient (*r*bs) was used to estimate the strength of association in comparisons between binary and continuous variables. Bonferroni correction was applied where multiple comparisons were made. A two-tailed *p*-value < 0.05 was considered statistically significant for all analyses. Statistical analyses were conducted using Statistica 11.0 (StatSoft Inc., Tulsa, OK, USA) and IBM SPSS Statistics v27 (IBM Corp., Armonk, NY, USA), with data visualization performed using GraphPad Prism 10 (GraphPad Software Inc., San Diego, CA, USA).

## 3. Results

### 3.1. Characteristics of the Studied Population

A total of 79 patients undergoing elective cardiac surgery were enrolled in the study (Table 1).

Preoperative serum IGFALS levels did not show significant correlations with age, BMI, CCI, or serum NT-proBNP. A modest correlation was observed between preoperative troponin I (TnI) levels and IGFALS measured at 24 h postoperatively (*r*^2^ = 0.17, *p* = 0.05), though this relationship was not sustained at later time points (7 days and 3 months post-surgery). Baseline or postoperative IGFALS levels were not significantly associated with hemoglobin A1c (HbA1c) or a pre-existing diagnosis of diabetes.

### 3.2. Changes of IGFALS and IGF After Surgery

Serum IGFALS levels demonstrated significant deviations from presurgical baseline across the perioperative period (KS[4;276] = 11.58, *p* = 0.009) at all postoperative time points (Figure 1A). The nadir occurred at t_7d_, with the strongest effect size (r_bs_ = −0.56). Reductions were also evident at t_24h_ (r_bs_ = −0.28) and t_3m_ (r_bs_ = −0.36). By three months, IGFALS levels showed partial recovery but remained below presurgical values (Figure 1A). In contrast, IGF-1 levels remained stable across all perioperative time points, showing no significant variability (KS[4;137] = 1.62; *p* = ns) (Figure 1B). Importantly, IGFALS levels at each postoperative time point showed strong correlations with baseline serum levels. (Figure 1C).

### 3.3. Postsurgical Clinical Recovery and IGFALS Dynamics

There were no correlations between IGFALS concentrations and intraoperative factors such as the duration of anesthesia, surgery, cardiopulmonary bypass, estimated blood loss, or the volume of crystalloids or blood products administered (data not shown). Serum IGFALS levels were not different between CABG and other types of heart surgery (Table 1). Significant differences in IGFALS concentrations post-surgery were observed among patients who received acetylsalicylic acid or over 1000 mg of acetaminophen, but only at 24 h sampling time (Figure 2A,B). Ketorolac, corticosteroids, opioids, and benzodiazepines did not affect post-operative IGFALS serum levels.

Both preoperative IGFALS and IGFALS at 24 h were significantly associated with APACHE II scores only at 48 h (IGFALS_preop_ vs. APACHE_48h_: KW(17;73) = 29.0, *p* = 0.034 and IGFALS_24h_ *vs* APACHE_48h_: KW(16;68) = 26.57, *p* = 0.047. No significant correlations were observed between IGFALS and APACHE II scores at later time points.

### 3.4. Relationship Between Perioperative Serum IGFALS and Serum Markers for Inflammation and Tissue Destruction

No meaningful associations were identified between IGFALS and the alarmins Hsp-60 or HMGB-1 at all sampling times. Correlations between IGFALS and classical inflammatory markers (IL-6 and IL-8) yielded several weak or inconsistent associations except t_24h_ time point (Figure 3).

### 3.5. Relationship Between Perioperative IGFALs and Functional Immunological Responses

Serum IGFALS demonstrated a moderate positive correlation with IL-6 production by whole blood in response to in vitro H3N2 stimulation at 24 h (*r* = 0.46, *p* = 0.019) and 7 days post-surgery (*r* = 0.43, *p* = 0.019). Baseline IGFALS levels were strongly correlated with baseline IL-6 production by isolated monocytes in response to in vitro H3N2 stimulation, which correlated with serum IGFALS levels at baseline (*r* = 0.51, *p* = 0.019) and 3 months (*r* = 0.607, *p* = 0.037). Production of cytokines by isolated monocytes and whole blood in response to LPS showed no statistically meaningful correlations.

### 3.6. Influence of Known Factors Affecting IGFALS Expression During the Perioperative Period

Growth hormone (GH), a known regulator of IGFALS expression, showed no significant correlation with IGFALS serum levels at any of the sampled time points, indicating that perioperative GH variability is unlikely to explain the observed changes in IGFALS (data not shown) [18,22,26].

IGFALS expression may be negatively regulated by interleukin-1β (IL-1β) and the suppressor of cytokine signaling 3 (SOCS3) pathway [26]. Although IL-1β mRNA was upregulated in monocyte transcriptomic data (Figure 4A), serum IL-1β protein levels remained below the detectable threshold throughout the study period (data not shown). SOCS3 gene expression was significantly elevated at 24 h and 7 days postoperatively but returned to lower levels by three months (Figure 4B), supporting a possible mechanistic link between immune signaling and sustained IGFALS suppression.

### 3.7. Changes of IGFALS with Acute Kidney Injury, Heart Failure, and Neurodegeneration Surrogates

Emergence of postoperative acute kidney injury was not associated with significant differences in IGFALS levels at baseline, 24 h, or 7 days. A modest correlation was observed between IGFALS levels at 24 h and preoperative troponin I levels (*r*^2^ = 0.17, *p* = 0.05), suggesting a potential association with subclinical myocardial injury. Presurgical IGFALS levels did not correlate with NT-proBNP levels at any time point.

Serum surrogates of neuronal insult demonstrated few significant associations with IGFALS levels (Figure 5). Most importantly, amyloid-β1 42 was significantly associated with IGFALS at baseline and 24 h post-surgery (Figure 5).

### 3.8. Correlation of IGFALS and Clinical Outcome

Serum IGFALS levels showed no significant correlation with the length of stay in the intensive care unit or total hospital stay. However, patients discharged home at 28 days had higher IGFALS levels both preoperatively and at 24 h post-surgery, compared to those discharged to a healthcare facility (Figure 6A). Similar associations were observed at the 3-month follow-up, where IGFALS levels continued to distinguish between patients with full functional recovery versus those requiring extended care (Figure 6B).

The incidence of cerebrovascular accidents (n = 10) did not significantly impact IGFALS levels at any measured time point. Due to the low number of events, no meaningful conclusions could be drawn regarding the association between IGFALS dynamics and other adverse outcomes, including postoperative deep vein thrombosis (n = 3), pulmonary embolism (n = 2), and mortality (n = 0).

## 4. Discussion

Our study demonstrates that elective cardiac surgery is associated with sustained depletion of serum IGFALS levels, persisting for up to three months postoperatively. Although a trend toward recovery was observed at the three-month mark, levels remained significantly below baseline, suggesting incomplete normalization in some patients. In contrast, IGF level showed no significant heterogeneity. These findings are consistent with prior observations in sepsis, where IGFALS levels declined during acute illness and eventually recovered in most, but not all, individuals [23,40,49]. Concomitantly, a previous study also reported only partial recovery of IGFALS at 30 days post-insult, reinforcing the possibility of prolonged suppression in a subset of patients [37].

Our data indicate that this depletion is more likely attributable to the effects of surgery itself rather than pre-existing cardiac conditions, as supported by the absence of correlation with NT-BNP and the lack of differences across types of cardiac procedures [37]. The weak correlation of IGFALS with troponin I at 24 h may point to a potential relationship with myocardial injury [50]. Considering that IGFALS is a matrix-associated protein, we expected some correlations and elevated IGFALS levels released from damaged tissues [17,51]. The lack of a relationship between serum IGFALS release and alarmin suggests that IGFALS is tied to myocardial injury. Consequently, its utility as a biomarker may be limited to cardiac surgery contexts. Lack of correlation between IGFALS and traditional inflammatory markers (e.g., IL-6, IL-8, IL-1β) suggests that IGFALS is not another simplistic general marker of inflammation. Previously, we showed that patient C-RP is elevated in the aftermath of cardiac surgery, suggesting theoretically inverse correlations between CRP and IGFALS [32]. This correlation was not tested in this study. Future studies could compare the post-operative IGFALS dynamic to cardiac surgery specific and generalized inflammation markers.

The observation that acetaminophen and aspirin influenced IGFALS levels is intriguing as little is known about control of IGFALS expression [2,36,52]. Both acetylsalicylic acid and acetaminophen act on the cyclooxygenase pathway, particularly COX-1, implying a shared mechanism potentially affecting IGFALS expression [53]. Acetaminophen, in particular, has been shown to impact epigenetic regulation, which may explain its long-term influence on IGFALS dynamics [52]. Interestingly, neither growth hormone (GH) nor IL-1β, both known regulators of IGFALS expression, were likely contributors to this postoperative depletion. Given that the duration of depletion exceeds the protein’s half-life and cannot be explained by fluid shifts alone, a transcriptional or translational suppression appears more plausible [15]. One plausible mechanism involves postoperative cellular polarization, which may elevate SOCS3 expression, a known suppressor of IGFALS transcription [32]. The strong correlation between baseline IGFALS levels and all subsequent post-surgical time points further supports a regulatory rather than an injury-driven process. SOCS3 expression followed a temporal pattern that may reflect immune reprogramming during postoperative recovery, raising the prospect that depletion of IGFALS is part of generalized post-surgical recovery [11,32,54]. IGFALS levels showed correlations with immune responsiveness to viral stimulation. This finding can be linked to the demonstrated in vitro IGFALS role in antiviral immunity [2,5,11]. However, more studies are needed to confirm this association. Also, IGFALS in the study by Xu et al was part of the intracellular response, while we measured serum levels. This is most likely because surgery is sterile inflammation.

The translational impact of our finding needs to be assessed, as several authors suggest a connection between IGFALS and adverse perioperative outcomes [21,23,24,30,34,36,37,40]. Our findings raise the possibility that IGFALS may be linked to postoperative neurocognitive dysfunction, as seen with other insulin-regulatory proteins [3,43]. However, the directionality of these relationships remains unclear. IGFALS alterations may reflect underlying metabolic or neurodegenerative conditions rather than causing them [31,33]. Given IGFALS’s role in glucose regulation, the link to progressive neurodegeneration in diabetic patients warrants further exploration [1,8,42]. Lower preoperative and early postoperative IGFALS levels were associated with delayed recovery at both 28 days and three months. This aligns with prior evidence linking IGFALS to non-valvular atrial fibrillation, heart failure, and recovery trajectories [37,38]. While speculative, these findings may reflect underlying impairments in glycemic control or immune resolution. Although the sample size of patients discharged to rehabilitation was small, the trend is consistent with earlier work suggesting IGFALS as a marker of recovery potential [3,30,41].

Our study has several advantages. The longitudinal design of our study helps mitigate this concern by anchoring changes to a presurgical baseline. High care standardization reduces the heterogeneity of certain confounders. IGFALS was measured using an established technique.

This pilot study has several significant limitations. The relatively small cohort limited our ability to assess complex multivariable associations, particularly with neurodegeneration outcomes [42]. The low rate of perioperative complications also constrained subgroup analyses. The adverse effects were extracted from the EMR and subjected to bias. Because all patients underwent elective cardiac surgery with cardiopulmonary bypass, a major physiological insult, generalizability to other surgical populations remains uncertain. That said, the lack of IGFALS differences across surgical subtypes may suggest broader applicability. Prior studies have also found minimal clinical distinction between on- and off-pump procedures [30,55]. In some cases, undetectable IGFALS levels may have been due to technical issues or genetic mutations affecting antibody recognition, both potential areas for future investigation. Additionally, pre-existing IGFALS abnormalities—whether due to chronic illness, gene mutations, or altered expression—could confound our findings [2,12,35]. As a single-center study, the findings may reflect specific institutional practices. These shortcomings are typical for a first-of-this-kind exploratory study.

Although the findings presented are novel, it is essential to replicate the study to confirm these results. The current research should provide sufficient data for power calculations, particularly if multivariate analysis of perioperative factors is incorporated in future investigations. Additionally, comparison with other surgical procedures is necessary to determine whether changes in serum IGFALS are unique to cardiac surgery or applicable across various surgical contexts.

## 5. Conclusions

Serum IGFALS levels are significantly depleted following cardiac surgery and remain suppressed for up to three months. Certain medications appear to modulate perioperative IGFALS levels. More pronounced depression in serum IGFALS is associated with poorer recovery outcomes. These findings suggest IGFALS may serve as a biomarker of postoperative recovery.

## Figures and Tables

**Figure 1 cimb-47-00581-f001:**
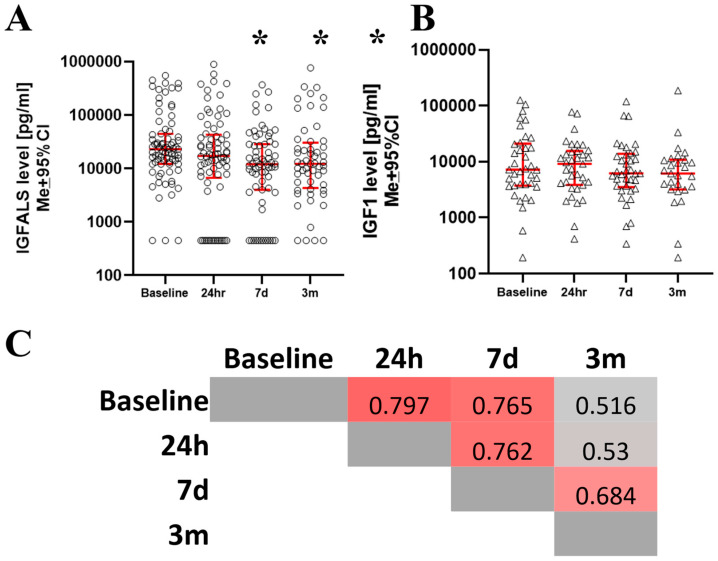
The serum level of IGFALS declined after surgery in absolute (**A**) and relative terms (**B**). However, IGF-1 serum levels remained unchanged. Correlation between postsurgical serum IGFALS correlates with all postoperative times (**C**). * m denotes where the difference between baseline and subsequent time points reaches statistical significance (*p* < 0.05).

**Figure 2 cimb-47-00581-f002:**
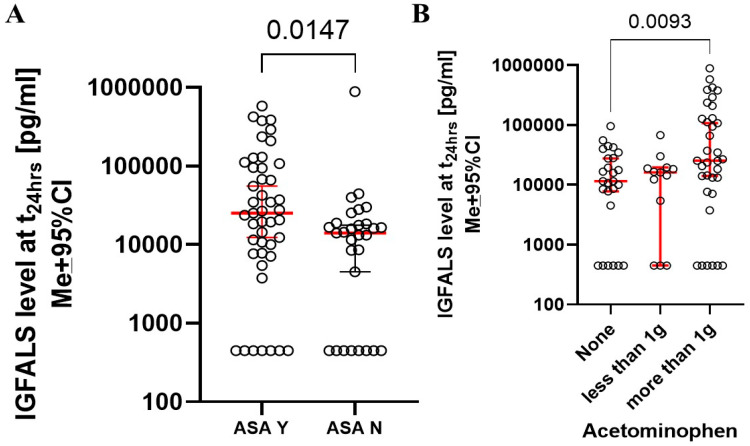
Perioperative intake of acetylsalicylic acid (**A**) and acetaminophen (**B**) resulted in increase of perioperative serum IGFALS levels.

**Figure 3 cimb-47-00581-f003:**
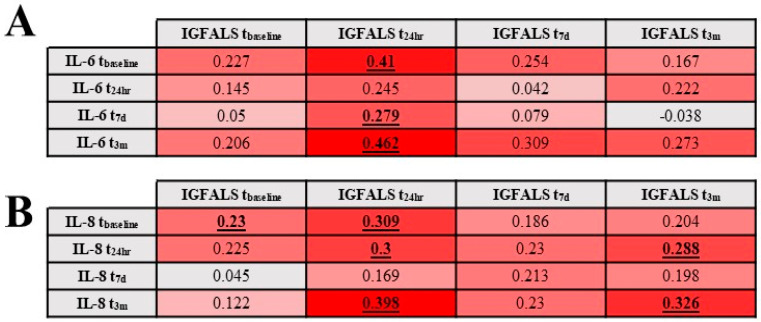
Correlation between IL-6 (**A**) and IL-8 (**B**) and serum levels of IGFALS. Underlined and bold numbers are statistically significant correlations with *p* < 0.05.

**Figure 4 cimb-47-00581-f004:**
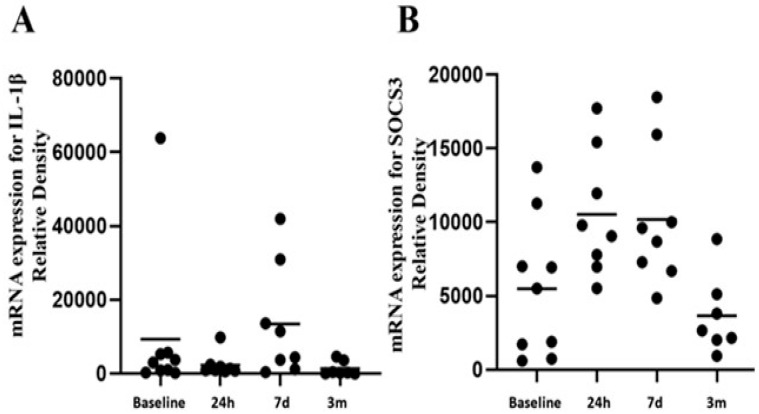
Analysis of the transcriptome revealed no changes in mRNA for IL-1β (**A**) of SOCS3 at t_3m_ despite its elevation at t_24h_ and t_7d_ (**B**) in peripheral monocytes.

**Figure 5 cimb-47-00581-f005:**
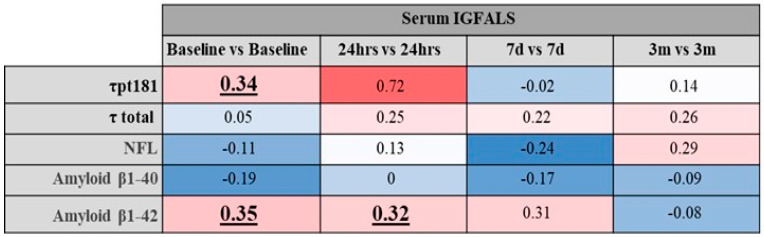
Correlation between IGFALS and markers of neuroinjury and neurodegeneration. Underlined and bold numbers are statistically significant correlations with *p* < 0.05.

**Figure 6 cimb-47-00581-f006:**
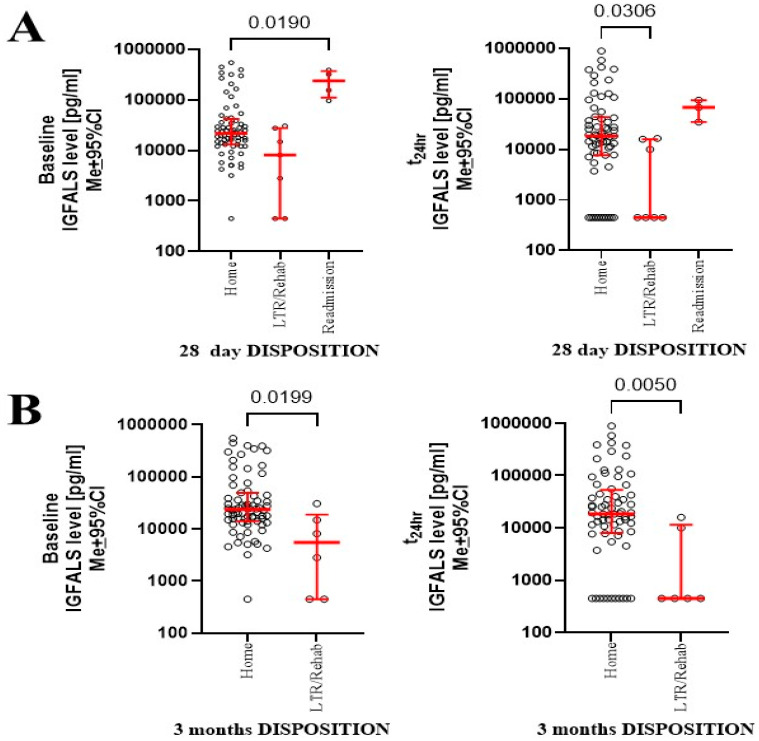
Outcomes at 28 days (**A**) and 3 months (**B**) were linked to serum levels of IGFALS and 4-month sampling points.

**Table 1 cimb-47-00581-t001:** Determinants of baseline IGFALS.

**DEMOGRAPHICS (n = 94)**	
**AGE [X ± SD]**	**63.2 ± 11.08**
**OVER 60 [%]**	**68.5%**
**GENDER**	
**MALE [%]**	**70.7%**
**FEMALE [%]**	**29.3%**
**NOT REPORTED [%]**	**0%**
**RACE**	
**WHITE [%]**	**89.1%**
**OTHER/ASIAN/BLACK/HISPANIC LATINO/UNKNOWN [%]**	**10.9%**
**PRE-EXISTING CONDITIONS**	
**BMI**	**27.7 ± 5.73**
**CHARLESTON COMORBIDITY INDEX [X ± SD]**	**3.9 ± 2.12**
**ACS/MI [%]**	**83.7%**
**CHF [%]**	**81.5%**
**PVD [%]**	**88%**
**CVA/TIA [%]**	**87%**
**DEMENTIA [%]**	**0%**
**COPD [%]**	**93.4%**
**DM [%]**	**75%**
**ANESTHESIA AND SURGERY DATA**	
**DURATION OF ANESTHESIA; MEAN ± SD [MIN]**	**376.9 ± 102.21**
**DURATION OF SURGERY; MEAN ± SD [MIN]**	**267.4 ± 97.35**
**DURATION OF CARDIOPULMONARY BYPASS; MEAN ± SD [MIN]**	**130.4 ± 64.49**
**CORONARY ARTERY BYPASS SURGERY; NO.**	**44**
**MITRAL VALVULOPLASTY AND REPLACEMENT; NO.**	**15**
**AORTIC VALVULOPLASTY AND REPLACEMENT; NO.**	**22**
**AORTIC ANEURYSM REPAIR; NO.**	**4**
**OTHERS; NO**	**7**
**TRANSFUSIONS DURING SURGERY**	
**PACKED RED BLOOD CELLS, MEAN (IQ25; IQ75) [ML]**	**159.8 [0; 0]**
**FRESH FROZEN PLASMA, MEAN (IQ25; IQ75) [ML]**	**154.9 [0; 0]**
**TOTAL CRYSTALLOID DURING SURGERY[ML]**	**97.3 ± 186.92**
**CLINICAL CARE DURING 24 H POST-SURGERY**	
**PACKED RED BLOOD CELLS, MEAN; (IQ25; IQ75) [ML]**	**19.8 [0;0]**
**FRESH FROZEN PLASMA, MEAN; (IQ25; IQ75) [ML]**	**0 [0; 0]**
**PERIOPERATIVE MEDICATIONS**	
**CORTICOSTEROID ADMINISTRATION (% OF ALL CASES)**	**1.1%**
**KETOROLAC ADMINISTRATION (% OF ALL CASES)**	**5%**
**ACETAMINOPHEN ADMINISTRATION (% OF ALL CASES)**	**69.6%**
**ACETYLSALICYLIC ACID ADMINISTRATION**	**67.4%**
**OPIOIDS ADMINISTRATION**	**90.1 ± 25.08**
**BZD ADMINISTRATION**	**3.5 ± 1.65**
**ICU STAY**	
**APACHE SCORE AT 1 H, MEAN ± SD**	**16.4 ± 5.83**
**APACHE SCORE AT 24 H, MEAN ± SD**	**9.2 ± 4.74**
**APACHE SCORE AT 48 H, MEAN ± SD**	**9.0 ± 4.28**
**OUTCOME AT 28 DAYS**	
**LOS ICU**	**9.3 ± 42.97**
**LOS HOSPITAL**	**10.5 ± 22.1**
**DVT**	**3.3%**
**PE**	**0%**
**CVA**	**14.1%**
**DISCHARGED/IN THE HEALTHCARE FACILITY/EXPIRED**	**88%/6.5%/4.3%**

## Data Availability

The datasets used and/or analyzed during the current study are available from the corresponding authors upon reasonable request.

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
