# Peer review of "Depletion of IGFALS Serum Level up to 3 Months After Cardiac Surgery, with Exploration of Potential Relationships to Surrogates of Organ Failures and Clinical Outcomes"

_cimb, 2025, doi:10.3390/cimb47080581_

Round 1

Reviewer 1 Report

Comments and Suggestions for Authors

Dear authors!

I have reviewed the following article,, Depletion of IGFALS serum level in the aftermath of cardiac 2 surgery – longitudinal surveillance up to 3 months of 79 pa- 3 tients undergoing an elective cardiac surgical procedure with 4 analysis of contributing factors and its link to surrogates of organ failures and clinical outcomes,,

First of all, congratulations for your work and thank you for the opportunity to collaborate in this project. In the following, I will analyze and add my observations regarding this manuscript

Abstract

The abstract is very complex, informative, but too long for an abstract. It is recommended not to exceed 250 words. This abstract is over 500 words, which makes it difficult to read quickly.

A clear structure: Background, Methods, Results, Conclusions, would greatly help in the rapid and efficient understanding of the abstract of this article. This approach helps the reader to quickly identify the context, methodology, results, and clinical relevance of the study. There is too much methodology, not every biomarker should be described in detail in the abstract, or all statistical values.

Introduction

It is scientifically sound, but could be improved statistically and in terms of clarity. There are some paragraphs that are too long, which are not easy to read. The purpose of the study should be more concise and clearly positioned

Materials and methods

The text is well-written, scientifically sound. This chapter does not require any changes.

Results

Although the results are promising and well presented, the relatively small sample size and the marginal statistical significance of some comparisons suggest that these data should be interpreted with caution. The lack of a statistically powerful analysis and the absence of adjusted multivariate models limit the strength of the conclusions, . Validation of the findings in a larger independent cohort would strengthen the clinical relevance of the observations regarding IGFALS.

Discussion

The discussion is well structured, balanced between data and interpretation, and offers multiple directions for future research. IGFALS is proposed not only as a potential marker of postoperative recovery, but also as a gateway to a broader field: the interaction between metabolism, inflammation, and neurodegeneration.

The limitations of the study are well highlighted.

Author Response

Response to Reviewer 1 Comments

1. Summary

2. Questions for General Evaluation

Reviewer’s Evaluation

Response and Revisions

Does the introduction provide sufficient background and include all relevant references?

Can be improved

Are all figures and tables clear and well-presented?

Yes

Is the research design appropriate?

Yes

Are the methods adequately described?

Yes

Are the results clearly presented?

Yes

Are the conclusions supported by the results?

Yes

3. Point-by-point response to Comments and Suggestions for Authors

Comments 1: Abstract

The abstract is very complex, informative, but too long for an abstract. It is recommended not to exceed 250 words. This abstract is over 500 words, which makes it difficult to read quickly.

A clear structure: Background, Methods, Results, Conclusions, would greatly help in the rapid and efficient understanding of the abstract of this article. This approach helps the reader to quickly identify the context, methodology, results, and clinical relevance of the study. There is too much methodology, not every biomarker should be described in detail in the abstract, or all statistical values.

Response 1: We agreed with the suggestion. Abstract was dramatically simplified

Comments 2: Introduction

It is scientifically sound, but could be improved statistically and in terms of clarity. There are some paragraphs that are too long and are not easy to read. The purpose of the study should be more concise and clearly positioned.

Response 2: Agree. We simplified, shortened, and adjusted the introduction. The repetitive sentences were removed. Aim of the study was contextualized and clearly stated

Comments 3: Materials and methods

The text is well-written, scientifically sound. This chapter does not require any changes.

Response 3: Thank you

Comments 4: Results

Although the results are promising and well presented, the relatively small sample size and the marginal statistical significance of some comparisons suggest that these data should be interpreted with caution. The lack of a statistically powerful analysis and the absence of adjusted multivariate models limit the strength of the conclusions, . Validation of the findings in a larger independent cohort would strengthen the clinical relevance of the observations regarding IGFALS.

Response 4: We couldn’t agree more with the reviewers. We have incorporated these comments into the results, discussion, and introduction. Our primary aim was to demonstrate that this is the first-of-its-kind exploratory and pilot study.

Comments 5: Discussion

The discussion is well-structured, balanced between data and interpretation, and offers multiple directions for future research. IGFALS is proposed not only as a potential marker of postoperative recovery but also as a gateway to a broader field: the interaction between metabolism, inflammation, and neurodegeneration.

The limitations of the study are well highlighted.

Response 5: Thank you. We added a more holistic view on IGFALS but also stress the limitations

Reviewer 2 Report

Comments and Suggestions for Authors

Thank you for having the opportunity to read the article entitled "Depletion of IGFALS serum level in the aftermath of cardiac surgery – longitudinal surveillance up to 3 months of 79 patients undergoing an elective cardiac surgical procedure with analysis of contributing factors and its link to surrogates of organ failures and clinical outcomes". It is interesting but I would have some suggestions and some questions for the authors:

1. First of all, the title is too long. The title should be shorter and present just the main idea.

2. The abstract has too many details. It should not be mention in the abstract too many details, especially about the enrollment process etc.

4. In the results section, there are repeated results. In the text it should appear just a comment, and the results (numbers) should be presented in the figures or in a table.

5. What about CRP or neutrophil/lymphocyte ratio?

6. What special medication did the patients have? Could it influence the results?

7. Did the patients have septic events?

8. What about cathecolamine and cortisol levels? It would be interesting to investigate also those parameters.

Author Response

Response to Reviewer 2 Comments

1. Summary

2. Questions for General Evaluation

Reviewer’s Evaluation

Response and Revisions

Does the introduction provide sufficient background and include all relevant references?

Must be improved

Revised

Are all figures and tables clear and well-presented?

Can be improved

We changed the layout of specific tables and simplified Figure 1

Is the research design appropriate?

Can be improved

The materials and methods section was reviewed. Results were clarified.

Are the methods adequately described?

Must be improved

We addressed the methods section

Are the results clearly presented?

Must be improved

The section was rephrased

Are the conclusions supported by the results?

Must be improved

We slightly adjusted the discussion

3. Point-by-point response to Comments and Suggestions for Authors

Comments 1: First of all, the title is too long. The title should be shorter and present just the main idea.

Response 1: Thank you for your feedback. We agree and have shortened the title.

Comments 2: The abstract has too many details. It should not be mentioned in the abstract too many details, especially about the enrollment process, etc.

Response 2: Acknowledged. The abstract has been shortened to include only essential details. The revised version is submitted for the reviewer's consideration.

Comments 3: In the results section, there are repeated results. In the text it should appear just a comment, and the results (numbers) should be presented in the figures or in a table.

Response 3: We removed a lot of redundancy. Figure 1 was simplified. Thank you for this excellent suggestion

Comments 4: What about CRP or neutrophil/lymphocyte ratio?

Response 4: We considered both tests but lacked sufficient CBC differential data to calculate the ratio. Post-surgery CRP was measured in separate work, so we did not repeat it here.

Comments 5: What special medication did the patients have? Could it influence the results?

Response 5: The term “special medication” is ambiguous. Most patients received standard perioperative care, as discussed in the manuscript. While some may have taken immunological or IGFALS-impacting medications, numerous covariates must be considered. We were afraid that such complex multivariate analysis is not be appropriate for this kind of study.

Comments 6: Did the patients have septic events?

Response 6: None of the patients had sepsis.

Comments 7:  What about catecholamine and cortisol levels? It would be interesting to investigate also those parameters.

Response 7: We considered both as indicators of stress and physiological response, but cortisol is difficult to measure. ACTH testing may be premature for this pilot study. Catecholamines would be affected by pressor injections, and even measuring their metabolites in urine—the gold standard—would introduce too many confounders for this manuscript.

Round 2

Reviewer 2 Report

Comments and Suggestions for Authors

The authors responded to all of my questions. The article is suitable for publication.